# Effect of Serotonin (5-Hydroxytryptamine) on Follicular Development in Porcine

**DOI:** 10.3390/ijms25179596

**Published:** 2024-09-04

**Authors:** Yan Zhang, Yu Han, Rui Yang, Bo-Yang Zhang, Yan-Sen Zhao, Yue-Qi Wang, Dao-Zhen Jiang, An-Tong Wang, Xue-Ming Zhang, Bo Tang

**Affiliations:** State Key Laboratory for Zoonotic Diseases, College of Veterinary Medicine, Jilin University, Changchun 130062, China; z_yan22@mails.jlu.edu.cn (Y.Z.); yuh7012@126.com (Y.H.); ruiyang22@mails.jlu.edu.cn (R.Y.); zhangboyang199807@163.com (B.-Y.Z.); yszhao21@mails.jlu.edu.cn (Y.-S.Z.); yueqiw22@mails.jlu.edu.cn (Y.-Q.W.); jiangdz23@mails.jlu.edu.cn (D.-Z.J.); wangat9921@mails.jlu.edu.cn (A.-T.W.); zhangxuem@jlu.edu.cn (X.-M.Z.)

**Keywords:** 5-hydroxytryptamine, follicles, granulosa cells, estrogen, pig

## Abstract

5-Hydroxytryptamine (5-HT) is an inhibitory neurotransmitter widely distributed in mammalian tissues, exerting its effects through binding to various receptors. It plays a crucial role in the proliferation of granulosa cells (GCs) and the development of follicles in female animals, however, its effect on porcine follicle development is not clear. The aim of this study is to investigate the expression of 5-HT and its receptors in various parts of the pig ovary, as well as the effect of 5-HT on porcine follicular development by using ELISA, quantitative real-time PCR (qPCR) and EdU assays. Firstly, we examined the levels of 5-HT and its receptors in porcine ovaries, follicles, and GCs. The findings revealed that the expression of different 5-HT receptors varied among follicles of different sizes. To investigate the relationship between 5-HT and its receptors, we exposed the GCs to 5-HT and found a decrease in 5-HT receptor expression compared to the control group. Subsequently, the treatment of GCs with 0.5 μM, 5 μM, and 50 μM 5-HT showed an increase in the expression of cell cycle-related genes, and EdU results indicated cell proliferation after the 0.5 μM 5-HT treatment. Additionally, the expression of genes involved in E2 synthesis was examined after the treatment of granulosa cells with 0.5 μM 5-HT. The results showed that CYP19A1 and HSP17β1 expression was decreased. These results suggest that 5-HT might affect the development of porcine follicle by promoting the proliferation of GCs and inhibiting the synthesis of estrogen. This provides a new finding for exploring the effect of 5-HT on follicular development, and lays a foundation for further research on the mechanism of 5-HT in follicles.

## 1. Introduction

The precise regulation of mammalian ovarian development is crucial for female reproductive health. Females are born with a complete set of germ cells, which reside within functional units known as follicles. Follicles contain oocytes, as well as specialized somatic granulosa cells (GCs), both of which are essential for oocyte survival [1]. Follicular development, hormone secretion, and oocyte maturation are three key factors that support female reproductive and endocrine functions [2]. Follicular development is characterized by four stages: primordial follicle, primary follicle, secondary follicle, and mature follicle, each stage significantly impacting ovarian vitality and egg quality. The total number of follicles, which serves as the basic functional unit of the mammalian ovary, is determined early in life and depletion of this number leads to reproductive senescence [3]. Follicular development is intricately regulated by a myriad of endocrine, paracrine, and autocrine factors [4]. Thus, it is essential to provide an appropriate environment for follicles to ensure their normal development.

GCs, as the predominant cell type in mature follicles, are responsible for synthesizing a variety of hormones and growth factors that are crucial for follicle initiation and growth [5]. These cells also regulate the growth, differentiation, and maturation of both somatic cells within the follicle and the oocytes themselves, thereby playing a pivotal role in follicular development. Consequently, the proliferation of GCs is closely linked to follicular development. 

Among the steroid hormones, estradiol (E2) is the most representative [6]. In ovarian tissue, the primary source of estrogen is androgens, which originate from the secretion of endomembrane cells. Following this, androgens enter GCs and accumulate within these cells. Upon induction by follicle-stimulating hormone (FSH), GCs produce aromatase, an enzyme that catalyzes the conversion of androgens to estrogens [7]. E2, synthesized and secreted by GCs, contributes significantly to follicular development; for instance, mice that are unable to produce E2 due to aromatase deficiency are rendered infertile [8]. In the ovaries, estrogen promotes follicle formation, inhibits GC apoptosis, and regulates steroid synthesis.

5-Hydroxytryptamine (5-HT), commonly known as serotonin, is a neurotransmitter widely distributed throughout the central nervous system and gastrointestinal tract, possessing functionalities akin to hormones and growth factors [9]. An increasing body of researches indicate that 5-HT plays a vital role in various physiological and pathological processes during animal development [10]. Furthermore, 5-HT exerts its diverse functions through the activation of specific receptors, which comprise seven subfamilies and operate via various signaling pathways [11]. Recent studies have highlighted the potential role of 5-HT in follicular development [12]. Specifically, 5-HT has been shown to regulate steroid hormone secretion and to participate in follicle development. For example, 5-HT stimulates the secretion of E2 and progesterone by preovulatory follicles and GCs in hamsters [13], and modulates E2 secretion in rat follicles. Additionally, 5-HT enhances progesterone secretion by bovine granulosa lutein cells [14] and promotes the secretion of both E2 and progesterone by cultured human GCs [15]. Moreover, 5-HT is implicated in follicular and embryonic development; maternal exposure to 5-HT prior to pregnancy can induce placental inflammation and reduce hormone secretion, adversely affecting offsprings development [16]. Studies have demonstrated that follicle-stimulating hormone and epidermal growth factor can enhance the growth and E2 secretion of porcine preantral follicles, thereby creating favorable conditions for follicular development and maintaining morphological integrity [17]. However, it remains unclear whether porcine follicles contain 5-HT and whether this neurotransmitter exerts an effect on porcine follicular development. Consequently, this study employed EdU staining, ELISA, and qPCR techniques to investigate these aspects. The findings from this research may offer new insights into the mechanisms by which 5-HT influences porcine follicular development. The primary objective of this experiment was to explore the impact of 5-HT on follicular development in pigs.

## 2. Results

### 2.1. The Content of 5-HT in Follicles of Different Sizes

To investigate the relationship between 5-HT levels and follicular development, we analyzed the concentration of 5-HT in porcine follicles of varying sizes. Follicles were isolated and their follicular fluid extracted based on established protocols. The criteria for follicle grading are detailed in Table 1. The analysis revealed that the concentration of 5-HT varied significantly among the different follicle sizes. In the middle-sized follicles, the 5-HT concentration was approximately 20 ng/mL, representing the lowest level observed among the three follicle size categories. Conversely, large follicles exhibited the highest concentration of 5-HT, at about 50 ng/mL (Figure 1A). The ELISA results confirmed the presence of 5-HT in follicles of all sizes, with distinct variations in concentration correlating with follicle size.

These findings suggest that 5-HT may play a crucial role in the process of follicular development, highlighting its potential significance in ovarian function and health.

### 2.2. Expression of 5-HT Receptors in the Ovary, Follicles and Granulosa Cells in Porcine

5-HT exerts its biological effects primarily through binding to specific 5-HT receptors. To investigate the mechanisms underlying the action of 5-HT in the porcine ovary, we assessed the expression of various 5-HT receptors in the ovary, follicles, and GCs using qPCR.

Our results demonstrated that the expression levels of the 5-HT receptors varied significantly across different tissues. Notably, the 5-HT4 and 5-HT2A receptors exhibited the highest expression levels in the ovary, followed by 5-HT7 and 5-HT6 receptors. Conversely, 5-HT1A, 5-HT3A, and 5-HT5A receptors showed minimal expression (Figure 2H).

When examining the expression of 5-HT receptors across follicles of varying sizes, we found that the expression levels of 5-HT2A, 5-HT3A, 5-HT5A, and 5-HT6 were relatively consistent, with no significant differences detected among different follicle sizes. In contrast, the expression of 5-HT1A, 5-HT4, and 5-HT7 receptors varied by follicle size. Specifically, the expression of 5-HT1A was the highest in large follicles, while the 5-HT4 receptor exhibited the lowest expression in this size category (Figure 1B–H).

Overall, these findings indicate that the expression of 5-HT and its receptors differs not only among various follicle sizes but also across different regions of the ovary. This suggests that 5-HT may exert diverse physiological effects through its interaction with distinct receptors, depending on the specific ovarian context.

### 2.3. Effect of 5-HT on Oocyte Maturation

To investigate the impact of 5-HT on porcine oocyte maturation, we supplemented the oocyte culture medium with varying concentrations of 5-HT. Specifically, the experimental group received 500 μM and 1000 μM of 5-HT, while the control group was maintained without treatment. After 48 h of culture, we assessed the rate of first polar body ejection as an indicator of oocyte maturation.

The results indicated that there was no significant difference in the rate of first polar body ejection between oocytes treated with 500 μM 5-HT and those in the control group. However, the addition of 1000 μM 5-HT resulted in a significant inhibition of first polar body ejection (Figure 3).

These findings suggest that high concentrations of 5-HT may adversely affect oocyte maturation by reducing the rate of first polar body ejection, indicating a potential inhibitory role of elevated 5-HT levels during this critical developmental process.

### 2.4. Effect of 5-HT on Porcine Granulosa Cells

GCs are the predominant cell type in mature follicles and play a crucial role in regulating follicular growth and development. To explore the regulatory mechanism of 5-HT on follicle growth, we utilized an in vitro cultured porcine granulosa cells (pGCs) model. We added varying concentrations of 5-HT to the pGCs culture medium to determine its effects. The qPCR results indicated that cell cycle-related genes, including Cyclin B1, Cyclin D1, and Cyclin E1, were significantly upregulated in the groups treated with 0.5 μM, 5 μM, and 50 μM of 5-HT compared to the control group (Figure 4A). To avoid potential cytotoxicity, 0.5 μM was selected as the optimal concentration for subsequent experiments. Additionally, the EdU results indicated that the positive cell rate of the control group was 6.05%, while that of the 5-HT group was 14.11%. This implies that 5-HT facilitated cell proliferation (Figure 4B).

These findings suggest that 5-HT promotes the proliferation of porcine granulosa cells, indicating its potential role in enhancing follicular growth and development through the regulation of cell cycle progression.

### 2.5. 5-HT Exposure Reduces the Expression of 5-HT Receptors

To investigate the effects of 5-HT on the expression of its receptors in porcine pGCs, we assessed the mRNA abundance of seven different 5-HT receptors: 5-HT1A, 5-HT2A, 5-HT3A, 5-HT4, 5-HT5A, 5-HT6, and 5-HT7. QPCR results revealed that the expression levels of all examined 5-HT receptors in the 5-HT-treated group were significantly lower compared to those in the control group (Figure 5).

These findings indicate that exposure to 5-HT leads to a downregulation of 5-HT receptor expression in porcine granulosa cells, suggesting a feedback mechanism that may modulate the responsiveness of these cells to 5-HT signaling.

### 2.6. 5-HT Exposure Inhibits E2 Synthesis in GCs

GCs in the ovary play a critical role in the secretion of E2, a hormone essential for follicle formation and the inhibition of granulosa cell apoptosis. To further elucidate the impact of 5-HT on E2 synthesis in GCs, we assessed the expression of E2 synthesis-related genes, specifically CYP19A1 (aromatase) and HSP17β1. The results showed that the mRNA levels of both HSP17β1 (Figure 6A) and CYP19A1 (Figure 6B) were significantly reduced in the 5-HT-treated group compared to the control group.

These findings suggest that exposure to 5-HT may inhibit E2 secretion by granulosa cells, potentially affecting follicular development and the overall reproductive function.

## 3. Discussion

5-HT is widely distributed in the female reproductive tissues of various animals, playing a crucial role in the development of mammalian follicles. Previous studies have identified 5-HT in human follicular fluid [18], as well as in the uterus and placenta of rats [19], and in mouse oocytes and cumulus cells [20]. 5-HT exerts its effects through binding to specific receptors, which are classified into seven subtypes in mammals. Notably, the expression of the 5-HT7 receptor has been observed in human GCs [21], and the 5-HT2A receptor has been identified in human oocytes [22]. Additionally, 5-HT2A, 5-HT2B [23], and 5-HT5A [24] receptors are expressed in the mouse female reproductive system. However, the expression of 5-HT and its receptors in pig ovaries, as well as their role in follicular development, remains largely unexplored.

In our study, we detected 5-HT in small, middle, and large follicles using ELISA, revealing that its expression levels vary according to follicle size, with the highest levels found in large follicles and the lowest in middle follicles. Furthermore, we analyzed the expression of seven 5-HT receptors in these three follicle types using qPCR. We observed variability in receptor expression across different follicle sizes, with 5-HT1A levels paralleling those of 5-HT. We also investigated the expression of these receptors in porcine ovaries and GCs of varying sizes. The mRNA expression of individual 5-HT receptors displayed distinct patterns in both the ovaries and GCs, with 5-HT2A and 5-HT4 showing high levels of expression in the ovaries. Previous studies have documented the presence of the 5-HT2A receptor in hamster oocytes [13] and in mouse cumulus cells [23]. Additionally, 5-HT has been shown to induce a Ca2+ surge in isolated stage II oocytes from warehouse rats [25], and the addition of 5-HT or its agonists to mouse cumulus-oocyte complexes caused an increase in Ca2+ and cAMP levels in cumulus cells. These findings suggest that serotonin receptors may potentially impact follicle development by binding to different receptors and exerting diverse effects. 

The aromatase enzyme, encoded by the CYP19A1 gene, is a key rate-limiting factor in the synthesis of E2 [26]. CYP19A1 is essential for E2 production in ovarian GCs; inhibiting CYP19A1 leads to reduced E2 secretion, while its overexpression enhances E2 production [27]. Dysregulation of CYP19A1 can result in impaired E2 synthesis and GC dysfunction. Our study found that exposure to 5-HT significantly inhibited the mRNA expression of CYP19A1, suggesting that 5-HT may suppress E2 synthesis in GCs. Interestingly, our results contradict some earlier studies that reported 5-HT stimulating E2 and progesterone secretion in pre-ovulatory follicles and GCs in hamsters [13], as well as E2 secretion in rat follicles [28] and cultured human GCs [15]. These earlier findings indicate that 5-HT may facilitate steroid hormone secretion and contribute to the maturation of oocytes. Conversely, other research has suggested that elevated 5-HT levels, particularly in the context of sleep deprivation, can inhibit serum estradiol concentrations in rats [29], aligning with our findings. We hypothesize that the proliferation of GCs may not be directly tied to E2 synthesis. The observed reduction in E2 synthesis could be attributed to increased concentrations of 5-HT. It appears that 5-HT exerts a concentration-dependent effect on E2 secretion: higher concentrations may inhibit E2 secretion and ovulation, while lower concentrations can promote E2 secretion and facilitate ovulation. This concentration-dependent effect of 5-HT parallels its influence on embryonic development, where varying concentrations produced differing outcomes. For instance, exogenous administration of 1 g/L of 5-HT inhibited mouse development, whereas lower concentrations (0.01 g/L and 0.10 g/L) slightly decreased embryonic development, and extremely low concentrations (0.0001 g/L) positively influenced development [30]. In conclusion, our findings contribute to a deeper understanding of the role of 5-HT and its receptors in the porcine reproductive system, particularly in the context of follicular development and estradiol synthesis. Further research is warranted to elucidate the complex mechanisms by which 5-HT influences reproductive functions in pigs and other mammals.

## 4. Materials and Methods

### 4.1. Collection of Tissues

Fresh sow ovaries were collected from a local abattoir, ensuring that sufficient quantities were obtained for the study. Any attached tissues, such as the ovarian frenulum, were carefully removed. The ovaries were then washed three times in normal saline containing 1% double antibody at 37 °C, followed by storage in a refrigerator at −80 °C for further analysis.

### 4.2. Follicles Classification

The collected fresh porcine ovaries were washed three times in a physiological saline solution supplemented with 1% penicillin-streptomycin after removing any adhering tissues. The ovaries were subsequently immersed in 75% alcohol for 30 s to ensure sterilization, then washed and stored in a 37 °C PBS water bath. Individual follicles were separated using autoclaved scissors and forceps, and classified based on criteria outlined in Table 1. 

### 4.3. Isolation and Culture of Porcine Granulosa Cells

Follicular fluid was extracted from the ovaries using a 10 mL injection syringe. Following centrifugation at 1000 rpm for 5 min, the supernatant was decanted, and the cell precipitate was washed multiple times with PBS solution. The liquid was filtered through a cell strainer, and the resulting pellet was resuspended in DMEM/F12 medium before being centrifuged again at 1000 rpm for 10 min. The final sediment was suspended in DMEM/F12 and inoculated into cell culture wells, where it was cultured in an incubator at 37 °C with 5% CO2 [31].

### 4.4. Oocytes Collection and In Vitro Culture

A total of 20 porcine ovaries were collected from the local abattoir and placed in 0.9% NaCl solution supplemented with penicillin. The samples were maintained at a temperature of 35–38.5 °C for 2–4 h during transport to the laboratory. A 10 mL syringe fitted with a 20-gauge needle was prepared for the aspiration of follicular fluid. Follicles measuring 3–6 mm in diameter were selected, and the fluid was aspirated through the syringe and allowed to sediment. A total of 150 cumulus-oocyte complexes (COCs) containing at least three layers of GCs were selected and washed twice with PBS supplemented with 10% fetal bovine serum. These COCs were then divided into three groups: a control group, a 500 μM 5-HT treatment group, and a 1000 μM 5-HT treatment group. The COCs were cultured in an in vitro maturation medium, first in maturation medium I for 22–24 h, followed by a transition to maturation medium II for an additional 42–44 h [32].

### 4.5. RNA Isolation and Quantitative Real-Time PCR

RNA was isolated from granulosa cells, follicles, and ovarian tissues using Trizol reagent. The RNA was subsequently reverse transcribed using the 1st-Strand cDNA Synthesis kit (Promega, Madison, WI, USA). QPCR was performed to analyze mRNA levels, utilizing primer sequences listed in Table 2. The reaction mixture consisted of 10 μL SYBR Green premix, 1 μL of each forward and reverse primer, 1 μL of cDNA, and 7 μL of RNase-free water. Reactions were conducted using a Roche Real-Time PCR Machine. Data were collected from at least three biological replicates. Relative transcription levels were normalized to the mRNA level of GAPDH and calculated using the 2-△△CT method.

### 4.6. ELISA

E2 present in the follicular fluid and medium supernatant was quantified using a porcine E2 ELISA Kit from Nanjing Jiancheng Bioengineering Institute (Nanjing, China), following the manufacturer’s protocol. The assay exhibited a coefficient of variation (CV) of less than 10% for intra-assay precision and less than 12% for inter-assay precision, with a sensitivity range from 20 to 6000 ng/L. The ELISA kit utilized monoclonal antibodies, demonstrating minimal cross-reactivity.

### 4.7. EdU Staining

GCs were treated with 5-HT and then seeded at a density of 1 × 10^3^ cells per well in 96-well plates. Following a 2-h incubation with EdU, the cells were washed twice with phosphate-buffered saline (PBS). They were subsequently fixed with 4% paraformaldehyde for 30 min, neutralized with 2 mg/mL glycine for 5 min, and permeabilized using 0.5% Triton X-100 for 5 min. The cells were then incubated with Click Additive Solution for 30 min, followed by three washes with 0.5% Triton X-100. After two washes with methanol, cells were stained with Hoechst for 30 min. Finally, the samples were observed under a fluorescence microscope for analysis.

### 4.8. Statistical Analysis

Statistical analyses were conducted using GraphPad Prism 6 software. All datasets were confirmed to follow a normal distribution and exhibited homogeneity of variance. One-way analysis of variance (ANOVA) was employed, complemented by the Newman-Keuls test for group comparisons. For comparisons between two groups, a paired Student’s *t*-test was utilized. Results are presented as mean ± standard error of the mean (SEM) from at least three independent experiments, with statistical significance denoted as * *p* < 0.05; ** *p* < 0.01.

## 5. Conclusions

The findings of this study demonstrate that porcine ovaries express 5-HT and its receptors, with exposure to 5-HT leading to a reduction in the mRNA expression of specific 5-HT receptors, particularly 5-HT2A and 5-HT7. By interacting with receptors on GCs, 5-HT influences the secretion of E2 and may play a regulatory role in follicular development. These results enhance the current understanding of how 5-HT modulates GCs function and suggest new avenues for exploring its impact on female reproductive potential. Furthermore, the process of follicular development appears to involve various signaling pathways that warrant additional investigation.

## Figures and Tables

**Figure 1 ijms-25-09596-f001:**
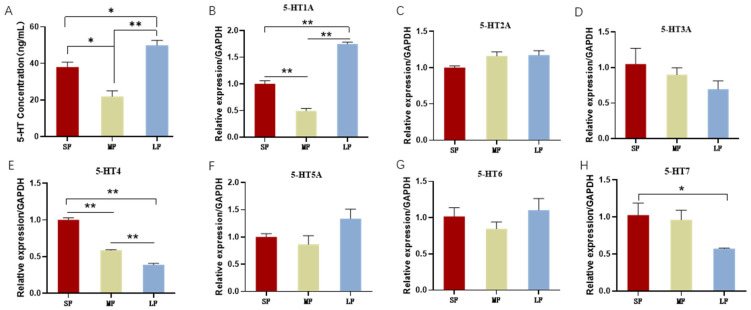
Effect of follicle size on 5-HT and 5-HT receptors. RNA was derived from the theca. SF: Small follicle; MF: Middle follicle; LF: Large follicle. 5-HT1A, 5-HT2A, 5-HT3A, 5-HT4, 5-HT5A, 5-HT6 and 5-HT7 are the seven receptors of 5-HT. (**A**) The content of 5-HT in follicles of different sizes was detected by ELISA. qPCR was used to detect the levels of 5-HT1A receptor (**B**), 5-HT2A receptor (**C**), 5-HT3A receptor (**D**), 5-HT4 receptor (**E**), 5-HT5A receptor (**F**), 5-HT6 receptor (**G**), and 5-HT7 receptor (**H**) in follicles of different sizes. * *p* < 0.05, ** *p* < 0.01.

**Figure 2 ijms-25-09596-f002:**
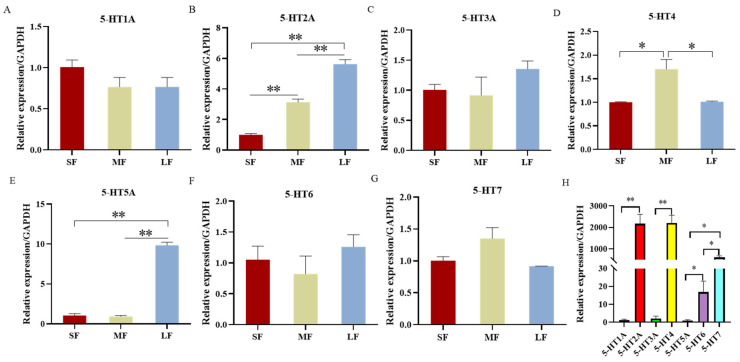
Expression of 5-HT receptors in pGCs and ovaries. RNA was derived from GCs and ovarian tissue. SF: Small follicle; MF: Middle follicle; LF: Large follicle. 5-HT1A, 5-HT2A, 5-HT3A, 5-HT4, 5-HT5A, 5-HT6 and 5-HT7 are the seven receptors of 5-HT. qPCR was used to detect the levels of 5-HT1A receptor (**A**), 5-HT2A receptor (**B**), 5-HT3A receptor (**C**), 5-HT4 receptor (**D**), 5-HT5A receptor (**E**), 5-HT6 receptor (**F**), and 5-HT7 receptor (**G**) in GCs of different sizes of follicles. (**H**) The expression of 5-HT receptors in ovary was detected by qPCR. * *p* < 0.05, ** *p* < 0.01.

**Figure 3 ijms-25-09596-f003:**
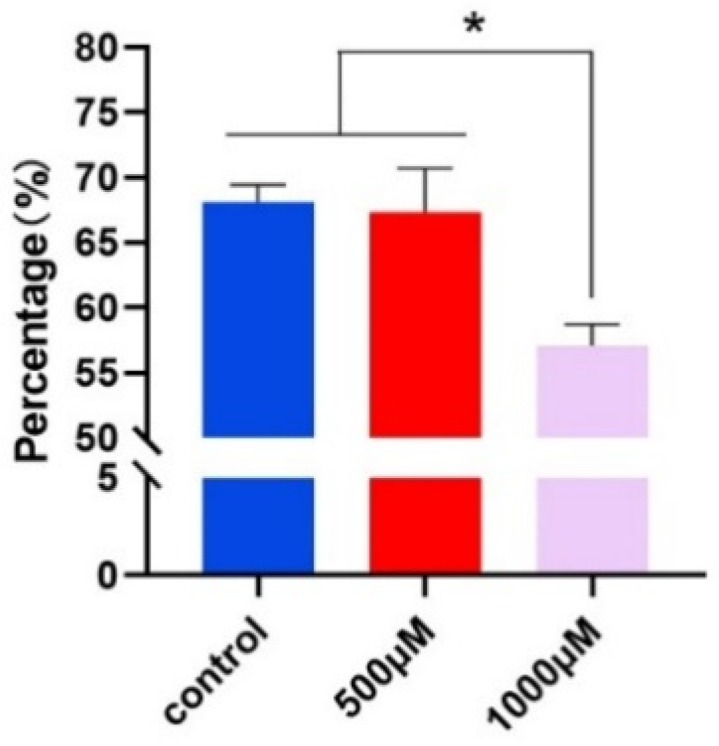
The effect of 5-HT on the first polar body of oocytes rate. The first polar body was treated with 0 μM, 500 μM, and 1000 μM 5-HT, respectively. * *p* < 0.05.

**Figure 4 ijms-25-09596-f004:**
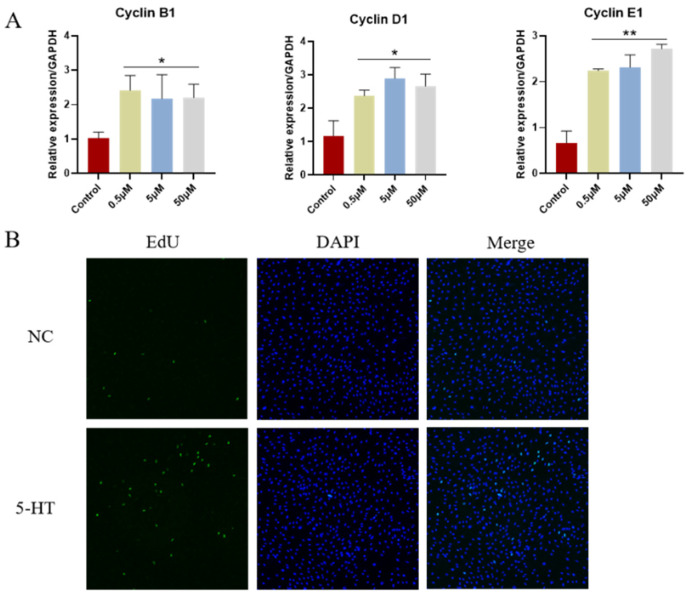
Effects of 5-HT exposure on pGCs. (**A**). q-PCR was used to detect the expression of cell cycle genes CyclinB1, CyclinD1 and CyclinE1. * *p* < 0.05, ** *p* < 0.01. (**B**). 5-HT exposure promotes proliferation of pGCs. The proliferation of pGCs determined by EdU assay. The chi-square test was conducted to evaluate the statistical difference in the proportion of positive cells between the control group (*n* = 215, positive = 13, negative = 202) and the experimental group (*n* = 317, positive = 45, negative = 272). The test revealed a significant difference (χ^2^ = 85.43, df = 1, *p* < 0.05), indicating that the proportion of positive cells varied significantly between the two groups. NC: Control group; 5-HT: 5-HT group.

**Figure 5 ijms-25-09596-f005:**
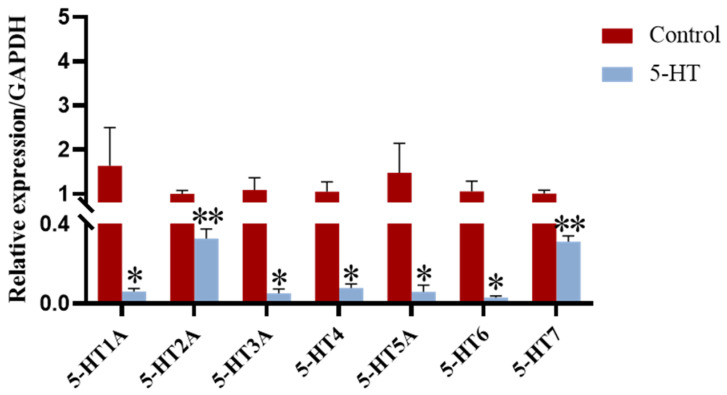
Effects of 5-HT exposure on 5-HT receptors in pGCs. 5-HT1A, 5-HT2A, 5-HT3A, 5-HT4, 5-HT5A, 5-HT6 and 5-HT7 are the seven receptors of 5-HT. * *p* < 0.05, ** *p* < 0.01.

**Figure 6 ijms-25-09596-f006:**
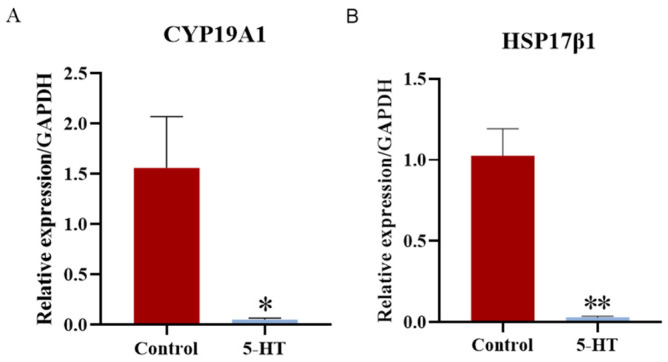
Effects of 5-HT exposure on E2 synthesis in pGCs. Changes in genes involved in E2 synthesis, including (**A**) Cyp19a1 and (**B**) HSP17β1, after 5-HT exposure. * *p* < 0.05, ** *p* < 0.01 vs. CON (*t*-test).

**Table 1 ijms-25-09596-t001:** Grading criteria for follicles. SF: Small follicle; MF: Middle follicle; LF: Large follicle.

Follicular Grading	Follicle Diameter
SF	2–3 mm
MF	3–5 mm
LF	5–6 mm

**Table 2 ijms-25-09596-t002:** Primers used for qRT-PCR.

Primer	Primer Pair Sequences (5′ to 3′)	Product Size (bp)
*Cyclin B1*	F: AATCCCTTCTTGTGGTTA	104
	R: CTTAGATGTGGCATACTTG	
*Cyclin D1*	F: TACACCGACAACTCCATCCG	224
	R: GAGGGCGGGTTGGAAATGAA	
*Cyclin E1*	F: AGAAGGAAAGGGATGCGAAGG	173
	R: CCAAGGCTGATTGCCACACT	
*CYP19A1*	F: TCCGCAATGACTTGGGCTAC	103
	R: GCCTTTTCGTCCAGTGGGAT	
*HSP17β1*	F: CCGAGCACCAAAGAGTGTTC	131
	R: TTATTGCGGGGTGGCAAGAG	
*5-HT1A*	F: TGCGTCCCCACCTCCTTCAAG	102
	R: GATGCCCAGCGTCTTCACAGTC	
*5-HT2A*	F: TCTTCCAGCGGTCCATCCACAG	129
	R: GGCACCACATCACCACGAACAG	
*5-HT3A*	F: TGACCTCCATCCGGCACTTCC	122
	R: CAGCACAGCCAGCAGGTAGATTC	
*5-HT4*	F: CGCATCTACGTGACAGCTAAGGAG	177
	R: CCAGCAGAGGCAGAAGCAACC	
*5-HT5A*	F: CCTCCTGTGGCTCGGCTACTC	80
	R: AGGTGCTGTTGTCGTGCTTGTC	
*5-HT6*	F: TTGGACTTGGACTCGGACTCAGG	134
	R: TTGAAGACGTTGACTGCGGTAGTG	
*5-HT7*	F: CCGTCAGGCAGAATGGCAAGTG	164
	F: CCGTCAGGCAGAATGGCAAGTG	

## Data Availability

The original contributions presented in the study are included in the article, further inquiries can be directed to the corresponding author.

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
