# Peer review of "Effect of Serotonin (5-Hydroxytryptamine) on Follicular Development in Porcine"

_ijms, 2024, doi:10.3390/ijms25179596_

Round 1
Reviewer 1 Report
Comments and Suggestions for Authors
Effect of 5-hydroxytryptamine on follicular development in porcine
Dear Authors,
The manuscript is interesting, and well prepared. Describies effect of serotonin on porcine follicular development additionally relation between expression of serotonin and its receptors in different parts of pig’s ovary.
Main problem in my opinion is subsection 2.2. Figures, Tables and Scheme, it is required to move those elements to Results subsections. Power of a test is very low it is important to use very conservative test in this case Scheffé’s or Bonferroni’s post-hoc test.
Below I add some suggestions helpful in this process:
Line 46
In text of manuscript is part of text: “…the main source of estrogen is androgens…”, should be: “…the main source of estrogen are androgens…”.
Line 71
Reference [17] is present in whole sentence two times, this first one can be deleted to avoid doubling of this reference in a row in one sentence.
Line 75
In last sentence purpose of conducted experiment can be added.
Line 81
In the text of manuscript is specified Table 2 which is localized in subsection 2.2. Figures, Tables and Scheme. Maybe better is to move this table to subsection 2.1.1. Levels of 5-HT secretion in different degrees of follicles, and numerate it as Table 1.
Lines 87-139
Generally tables are very rare separated from Results subsection, better is to movea all figures and tables to sub sections where they are mentioned with descriptions and delete subsection 2.2. Figures, Tables and Scheme.
Line 146
P-value is specified, but n=3, in this case p-value must be used.
Lines 155 and 158
P-value is specified, but n=3, in this case p-value must be used.
Lines 161 and 166
P-value is specified, but n=3, in this case p-value must be used.
Line 170
P-value is specified, but n=3, in this case p-value must be used.
Line 178
The same as in line 71. Reference [23] can be emphasized one time in this sentence, first can be deleted to avoid doubling references in one sentence.
Line 192
Reference (Terranova et al., 1990) is specified in the text, in this case [13] can be used. The same in line 193 and 197 (Amireault and Dubé, 2005), can be described as [23].
Line 224
(Koppan et al., 2004), can be specified as [15].
Line 248
Information about number of samples must be added to this subsection. In case of figures n=3 is specified, in this case that will gives very low power of a test. Maybe in following experiments will be important to increase number of samples minimally to 10. In this case better will be to use from statistical point of view more conservative post-hoc tests: Scheffé’s or Bonferroni’s.
Line 291
Information about normality of distribution and homogeneity test can be added. Newman-Keuls test was used, but in case of this analysis the most conservative tests are recommended: Scheffé’s or Bonferroni’s (n=3 in treatment).
Line 295
Same as in line 146, p-value must be used.
Lines 319-396
DOI links must be added.
Abbreviations of Journal’s name are required, where it is possible. From reference no. 3.
Author Response
The manuscript is interesting, and well prepared. Describies effect of serotonin on porcine follicular development additionally relation between expression of serotonin and its receptors in different parts of pig’s ovary.
Main problem in my opinion is subsection 2.2. Figures, Tables and Scheme, it is required to move those elements to Results subsections. Power of a test is very low it is important to use very conservative test in this case Scheffé’s or Bonferroni’s post-hoc test.
Response: Thank you for your comments. Based on your suggestions, Section 2.2, including figures, tables, and diagrams, has been moved to the Results subsection. We apologize for an error in the article. In the figure captions, the number of N was mistakenly written as 3, leading to an incorrect representation of the sample size in the experiment. We have corrected this mistake and selected the appropriate testing method.
Below I add some suggestions helpful in this process:
- Line 46
In text of manuscript is part of text: “…the main source of estrogen is androgens…”, should be: “…the main source of estrogen are androgens…”.
Response: Many thanks. We have revised as the comments of the reviewer in the manuscript.
- Line 71
Reference [17] is present in whole sentence two times, this first one can be deleted to avoid doubling of this reference in a row in one sentence.
Response: Thank you! We have removed a duplicate reference.
- Line 75
In last sentence purpose of conducted experiment can be added.
Response: Thank you for the advice. We added at the end:“The purpose of this experiment was to explore the effect of 5-HT on the development of porcine follicles .”
- Line 81
In the text of manuscript is specified Table 2 which is localized in subsection 2.2. Figures, Tables and Scheme. Maybe better is to move this table to subsection 2.1.1. Levels of 5-HT secretion in different degrees of follicles, and numerate it as Table 1.
Response: Thank you for the advice. 2.2. Figures, Tables and Scheme have been moved to the Results subsections. The levels of 5-HT secretion in follicles of different sizes are presented in Figure1A, so we did not add them to Table 2. Figure 1A shows the 5-HT content we measured from the same number of large, medium, and small follicles.
- Lines 87-139
Generally tables are very rare separated from Results subsection, better is to movea all figures and tables to sub sections where they are mentioned with descriptions and delete subsection 2.2. Figures, Tables and Scheme.
Response: Thank you for the advice. 2.2. Figures, Tables and Scheme have been moved to the Results subsection.
- Line 146
P-value is specified, but n=3, in this case p-value must be used.
Response: Many thanks for this valuable suggestion. We made a mistake in the description of the number of experimental samples, and we have corrected the errors in the manuscript.
- Lines 155 and 158
P-value is specified, but n=3, in this case p-value must be used.
Response: Many thanks for this valuable suggestion. We made a mistake in the description of the number of experimental samples, and we have corrected the errors in the manuscript.
- Lines 161 and 166
P-value is specified, but n=3, in this case p-value must be used.
Response: Many thanks for this valuable suggestion. We made a mistake in the description of the number of experimental samples, and we have corrected the errors in the manuscript.
- Line 170
P-value is specified, but n=3, in this case p-value must be used.
Response: Many thanks for this valuable suggestion. We made a mistake in the description of the number of experimental samples, and we have corrected the errors in the manuscript.
- Line 178
The same as in line 71. Reference [23] can be emphasized one time in this sentence, first can be deleted to avoid doubling references in one sentence.
Response: Thank you! We have revised in the manuscript.
- Line 192
Reference (Terranova et al., 1990) is specified in the text, in this case [13] can be used. The same in line 193 and 197 (Amireault and Dubé, 2005), can be described as [23].
Response: Thank you! We have revised in the manuscript.
- Line 224
(Koppan et al., 2004), can be specified as [15].
Response: Thank you! We have revised in the manuscript.
- Line 248
Information about number of samples must be added to this subsection. In case of figures n=3 is specified, in this case that will gives very low power of a test. Maybe in following experiments will be important to increase number of samples minimally to 10. In this case better will be to use from statistical point of view more conservative post-hoc tests: Scheffé’s or Bonferroni’s.
Response: Thank you for your comments. Information on the number of samples has been added to this subsection. We made a mistake in the expression of the number of samples in the experiment. We have completed the modification and selected the appropriate test methods.
- Line 291
Information about normality of distribution and homogeneity test can be added. Newman-Keuls test was used, but in case of this analysis the most conservative tests are recommended: Scheffé’s or Bonferroni’s (n=3 in treatment).
Response: Thank you for your suggestion. We made a mistake in the expression of the number of samples in the experiment. We have completed the modification and selected the appropriate test methods accordingly.
- Line 295
Same as in line 146, p-value must be used.
Response: Many thanks for this valuable suggestion. We made a mistake in the description of the number of experimental samples, and we have corrected the errors in the manuscript.
- Lines 319-396
DOI links must be added.
Abbreviations of Journal’s name are required, where it is possible. From reference no. 3.
Response: Thank you so much for your efforts to improve the quality of our manuscript. References of the DOI links has been added. The names of the journals were abbreviated whenever possible.
Reviewer 2 Report
Comments and Suggestions for Authors
The authors describe the effect of serotonin on the development of porcine follicular cells. The paper is well-written, and I have no concerns regarding the language. I recognize the significant effort that has gone into these studies, and I commend the authors for their work. However, as a reviewer, I must raise some questions that I believe are crucial to the overall evaluation of the research.
Comments 1: Article Title
I understand that the choice of the title is the authors' prerogative, and this comment is more of a suggestion than something essential for this paper. However, 5-hydroxytryptamine is known to a wider audience as serotonin. So maybe the title "Effect of serotonin (5-hydroxytryptamine) on follicular development in porcine" will be more informative for people who are not familiar with the chemical name of serotonin.
Comments 2: About follicle secretion (lines 78 to 86)
As far as I know, serotonin is a neurosecretory hormone and is not secreted or produced by follicular cells. I believe this is an accumulation of serotonin from the bloodstream, facilitated by SERT (serotonin membrane transporter). Moreover, the authors use follicular fluid as proof of secretion; once again, it is an accumulation, not a secretion. The level of secretion can be measured over time, such as Xng per Xh or similar. The described 20ng/mL is a concentration unit, not a secretion rate. I think this is a major mistake and should be considered a reason for a major revision of the experimental methodology described in this manuscript.
Comments 3: Description in figures
The figure legends do not have proper explanations. I know that SF, MF, and LF represent follicular stages, but that should be marked in the figure descriptions. The same is true for other figure descriptions; every abbreviation used in the figures should be expanded in the figure legends. This is the standard practice for scientific publications
Comments 4: Pro-Proliferative Effect of Serotonin on Follicles
Was any other method used to determine the pro-proliferative effect of serotonin on the follicles? From what I can see, only fluorescent images are presented. Were the cells with positive signals counted? If so, how many were counted? It would be a good idea to perform flow cytometry analysis to quantify how this proliferation changes. Currently, we only see fluorescent images, which do not provide any statistical data to support the authors' thesis that serotonin is pro-proliferative.
Author Response
The authors describe the effect of serotonin on the development of porcine follicular cells. The paper is well-written, and I have no concerns regarding the language. I recognize the significant effort that has gone into these studies, and I commend the authors for their work. However, as a reviewer, I must raise some questions that I believe are crucial to the overall evaluation of the research.
- Comments 1: Article Title
I understand that the choice of the title is the authors' prerogative, and this comment is more of a suggestion than something essential for this paper. However, 5-hydroxytryptamine is known to a wider audience as serotonin. So maybe the title "Effect of serotonin (5-hydroxytryptamine) on follicular development in porcine" will be more informative for people who are not familiar with the chemical name of serotonin.
Response: Thank you for your comments. We've changed the title to “Effect of serotonin (5-hydroxytryptamine) on follicular development in porcine.”
- Comments 2: About follicle secretion (lines 78 to 86)
As far as I know, serotonin is a neurosecretory hormone and is not secreted or produced by follicular cells. I believe this is an accumulation of serotonin from the bloodstream, facilitated by SERT (serotonin membrane transporter). Moreover, the authors use follicular fluid as proof of secretion; once again, it is an accumulation, not a secretion. The level of secretion can be measured over time, such as Xng per Xh or similar. The described 20ng/mL is a concentration unit, not a secretion rate. I think this is a major mistake and should be considered a reason for a major revision of the experimental methodology described in this manuscript.
Response: Thank you for your comments. We have revised the entire text according to your suggestions. In addition to being a neurotransmitter, 5-HT is also used as an autoactive substance. According to current literature, the sources of 5-HT mainly come from the following three pathways. First, enterochromaffin cells can synthesize and secrete 5-HT [1]. Second, neurons in the central nervous system can also synthesize and secrete 5-HT [2]. Third, other studies have reported that specific cells in certain organs can synthesize and secrete 5-HT. For example, there were early clues that granulosa cells might produce 5-HT [3]. However, in our experiments, we have no direct evidence that serotonin in follicular fluid is secreted by follicular cells, so your suggestion is very accurate.“The described 20ng/mL is a concentration unit, not a secretion rate.”Expressing it as “content” is more precise than “secretion level.” Therefore, we have revised the entire text according to your suggestions. For example, “Levels of 5-HT secretion in different degrees of follicles” was changed to “The content of 5-HT in follicles of different sizes” in line 80. Similar revisions were made to lines 82, 85, and 86.
References:
[1] Liu N, Sun S, Wang P, Sun Y, Hu Q, Wang X. The Mechanism of Secretion and Metabolism of Gut-Derived 5-Hydroxytryptamine. Int J Mol Sci. 2021 Jul 25;22(15). doi: 10.3390/ijms22157931.
[2] Mohammad-Zadeh LF, Moses L, Gwaltney-Brant SM. Serotonin: a review. J Vet Pharmacol Ther. 2008 Jun;31(3):187-99. doi: 10.1111/j.1365-2885.2008.00944.x.
[3] Dubé F, Amireault P. Local serotonergic signaling in mammalian follicles, oocytes and early embryos. Life Sci. 2007 Dec 14;81(25-26):1627-37. doi: 10.1016/j.lfs.2007.09.034.
- Comments 3: Description in figures
The figure legends do not have proper explanations. I know that SF, MF, and LF represent follicular stages, but that should be marked in the figure descriptions. The same is true for other figure descriptions; every abbreviation used in the figures should be expanded in the figure legends. This is the standard practice for scientific publications
Response: Thank you for your efforts to improve the quality of the manuscript. We added “SF: Small follicle; MF: Middle follicle; LF: Large follicle. 5-HT1A, 5-HT2A, 5-HT3A, 5-HT4, 5-HT5A, 5-HT6 and 5-HT7 are the seven receptors of 5-HT.” to the note in figure 1 and 2. Added “The first polar body was treated with 0μM, 500μM, and 1000μM 5-HT, respectively.” in the note of Figure 3.Added “5-HT1A, 5-HT2A, 5-HT3A, 5-HT4, 5-HT5A, 5-HT6 and 5-HT7 are the seven receptors of 5-HT.” in the note of Figure 5.
- Comments 4: Pro-Proliferative Effect of Serotonin on Follicles
Was any other method used to determine the pro-proliferative effect of serotonin on the follicles? From what I can see, only fluorescent images are presented. Were the cells with positive signals counted? If so, how many were counted? It would be a good idea to perform flow cytometry analysis to quantify how this proliferation changes. Currently, we only see fluorescent images, which do not provide any statistical data to support the authors' thesis that serotonin is pro-proliferative.
Response: Thank you! Using flow cytometry to determine cell proliferation is indeed an excellent method. However, we regret that this was not included in the initial design of our experiment.
In our paper, we employed two methods to assess the proliferation of GCs. First, we used the EdU assay to measure cell proliferation. The EdU assay is widely adopted in the field of cell proliferation detection due to its high sensitivity, the ability to be directly observed under a fluorescence microscope, ease of operation, short experimental time, and the advantage of not requiring DNA denaturation.
Furthermore, in addition to visually observing cell proliferation, we also examined cell proliferation at the molecular level. We measured the expression levels of cell cycle proteins Cyclin B1, Cyclin D1, and Cyclin E1. If the cells are proliferating, the expression of these cell cycle protein genes would increase. Therefore, through the EdU assay and qPCR analysis of cell cycle proteins, we confirmed that GCs were indeed proliferating.
Consequently, we believe that serotonin promotes the proliferation of granulosa cells. However, we did not provide a statistical count of cells showing positive signals, and we have supplemented this data in the modified manuscript notes.
Reviewer 3 Report
Comments and Suggestions for Authors
The authors evaluated the effect of 5-HT on follicular development. Although the subject is interesting in the field. This reviewer believes that the manuscript needs to be improved before being accepted.
Major comments:
This reviewer is not convinced that the authors can conclude what they did based on their results. The authors seem to be overinterpreting the results and/or inferring more than the data allows.
The authors conclude that 5-HT affects the secretion of E2. However, there is no data indicating changes in E2 levels. The authors evaluated the expression levels of 2 genes.
Lines 211-213. The authors indicate that proliferation and estradiol secretion were inhibited after exposure of GCs to 5-HT. I do not believe this is true based on the data and/or lack of evaluations. Consider including other evaluations and make sure the result descriptions and the conclusions are based on the results.
This reviewer does not believe that the authors can state that the secretion of estradiol was inhibited just based on decreased gene expression of 2 genes involved in the pathway. I would suggest including the measurement of estradiol levels to confirm and support the statement.
The introduction needs to be reorganized as the information is disjointed. In the first paragraph, the authors are talking about follicles. In the second one, the authors introduce the term follicles.
Why did the authors decide to perform an analysis of the effect of 5-HT on meiosis when they are interested in evaluating the role of 5-HT on follicle development?
It would be good to include a better explanation on the figure legend, so the reader understands if the evaluations were performed in GCs, follicles, ovary etc. Some figures lack of this information.
What is the rationale for using the concentrations in Figure 3? Concentrations seem pretty high keeping in mind the ones used in figure 4. Can the decreased polar body extrusion at the highest concentration be due to cytotoxicity? Please clarify.
Minor comments:
Line 40. Define GCs
Line 46. Which 3?
Line 75. On porcine cells…or in follicle development as that is what the authors stated they are interested in?
Line 77. Results section is duplicated
Line 79. Verify? That means there is already a correlation established and you are just confirming it? Can the authors please provide the reference or rewrite the sentence?
This reviewer does not see any relevance for Table 2.
Line 82. “which was the lowest in the whole follicle” What does it mean?
Line 94. No significant difference in the size of follicles? Are the authors indicating that they analyzed size differences in the follicles using statistical analysis or that there was no difference in gene expression levels in the different follicles (based on the size)?
Line 97. 5-HT and 5-HT1A… is it correct? Did the authors quantify the expression of 5-HT or the levels?
How long the COCs were exposed to 5HT? really important information is missing throughout the manuscript.
Line 114. Should the sentence be in the past tense?
GCs and pGC?
Line 119. Is “improved” the appropriate word to use?
Line 190. 5-H2A?
Line 201. Did the authors evaluate ovulation?
Lines 204-205 I do not believe it is possible to use GCs to investigate the regulatory mechanisms of 5-HT on follicle growth. Please, clarify
Line 257. How did the authors perform the follicle separation and measure of the follicles?
Line 258. Table 1?
Line 261. Did the authors wash the “follicular fluid” or the cells?
Line 266. Where does this RNA come from? It is not clear what tissue/follicle population the authors used. Please clarify.
Sections in the materials and methods can be rearranged to have a logical sequence.
Line 272. Rohche?
Line 276. 1x103? Superscript
Line 278. Treated with the fixation… fixed
Section 4.6 are the authors using a commercial kit? The authors need to clarify what are the reagents B, C, D, and E.
Comments on the Quality of English Language
English editing is needed as the document is poorly written with numerous grammatical and writing errors. It is hard to follow and confusing. Some examples are in the below.
Line 61. Folliculars
Line 78. Different degrees of follicles?
Line 108. Ma turation
Author Response
The authors evaluated the effect of 5-HT on follicular development. Although the subject is interesting in the field. This reviewer believes that the manuscript needs to be improved before being accepted.
Thank you so much for your constructive suggestions.
Major comments:
This reviewer is not convinced that the authors can conclude what they did based on their results. The authors seem to be overinterpreting the results and/or inferring more than the data allows.
Response: Thank you for your comments. From a certain perspective, you are correct that our results do not directly support the conclusion. In our experiments, the main findings after treating GCs with serotonin were: 1) GCs proliferation; 2) a decrease in E2-Synthesis related genes, suggesting a possible decrease in E2; and 3) a decrease in the expression of serotonin receptors in GCs. These results are primarily phenotypic and do not delve into the underlying mechanisms.
However, our conclusions were drawn based on well-established literature: GCs can synthesize E2 [1]; E2 can promote ovulation [2]; and serotonin exerts its effects through binding to serotonin receptors [3]. Based on these known relationships and our experimental results, we suggest that serotonin is negatively correlated with the secretion of E2 by GCs and may exert its effects through binding to serotonin receptors. As you mentioned, we should use more cautious language, adding “possibly” or “may” rather than making definitive statements. We have revised the conclusive sentences throughout the manuscript to make them more precise.
References:
[1] Edson MA, Nagaraja AK, Matzuk MM. The mammalian ovary from genesis to revelation. Endocr Rev. 2009 Oct; 30(6):624-712. doi: 10.1210/er.2009-0012.
[2] Richards JS, Pangas SA. The ovary: basic biology and clinical implications. J Clin Invest. 2010 Apr;120(4):963-72. doi: 10.1172/JCI41350.
[3] Tierney AJ. Structure and function of invertebrate 5-HT receptors: a review. Comp Biochem Physiol A Mol Integr Physiol. 2001 Apr;128(4):791-804. doi: 10.1016/s1095-6433(00)00320-2.
The authors conclude that 5-HT affects the secretion of E2. However, there is no data indicating changes in E2 levels. The authors evaluated the expression levels of 2 genes.
Lines 211-213. The authors indicate that proliferation and estradiol secretion were inhibited after exposure of GCs to 5-HT. I do not believe this is true based on the data and/or lack of evaluations. Consider including other evaluations and make sure the result descriptions and the conclusions are based on the results.
This reviewer does not believe that the authors can state that the secretion of estradiol was inhibited just based on decreased gene expression of 2 genes involved in the pathway. I would suggest including the measurement of estradiol levels to confirm and support the statement.
Response: Thank you for your comments. We completely agree with your perspective. If we want to study the effect of serotonin on the secretion of E2, directly measuring the concentration of E2 would indeed be the best approach, just as we did in our experiment using the ELISA method to measure the concentration of E2 in follicular fluid, which provides a more straightforward analysis. So we modified the description of the results and conclusion. The word“may”was added.
In fact, we originally designed the experiment in this way. However, due to the lockdown in China during the COVID-19 pandemic, the main experimenter, Han Yu, who was responsible for the majority of the experimental operations, was facing graduation and did not have enough time to complete the E2 detection experiments. Therefore,we could only provide data on the changes in E2 synthesis-related genes to support the possible changes in E2 secretion.
To further explain this, we have included some literature references that suggest that changes in E2 synthesis genes can, to a certain extent, reflect changes in E2 levels. Currently, since the main experimental personnel have already graduated, it would be quite difficult to supplement this data. We hope you can understand, and thank you for your patience!
We examined the effect of 5-HT on GCs proliferation by EdU and cell cyclin-related genes and showed that 5-HT promoted GCs proliferation. As far as known, Estradiol can be synthesized by GCs. The aromatase P450arom encoded by the CYP19A1 gene is the key rate-limiting enzyme in the E2 synthesis pathway [1]. HSP17β1 plays a central role in the homeostatic regulation of sex steroid hormones by converting weakly active E1 to highly active E2 [2]. So we expressed changes in E2 by examining the expression of both genes. I think adding measurement of estradiol levels will make our results more convincing, but this experiment was not designed at the beginning, and the main author has graduated. So we modified the description of the results appropriately. I hope you can understand.
References:
[1] Cheng, J. C.; Fang, L. L.; Yan, Y.; He, J. Y.; Guo, Y. J.; Jia, Q. Q.; Gao, Y. B.; Han, X. Y.; Sun, Y. P., TGF-β1 stimulates aromatase expression and estradiol production through SMAD2 and ERK1/2 signaling pathways in human granulosa-lutein cells. J Cell Physiol. 2021, 236, (9), 6619-6629. https://doi.org/10.1002/jcp.30305.
[2] Heinosalo T, Saarinen N, Poutanen M. Role of hydroxysteroid (17beta) dehydrogenase type 1 in reproductive tissues and hormone-dependent diseases. Mol Cell Endocrinol. 2019 Jun 1; 489: 9-31. doi: 10.1016/j.mce.2018.08.004.
The introduction needs to be reorganized as the information is disjointed. In the first paragraph, the authors are talking about follicles. In the second one, the authors introduce the term follicles.
Response: Thank you for your comments. We have made minor adjustments to the order of the sentences in the introduction
It would be good to include a better explanation on the figure legend, so the reader understands if the evaluations were performed in GCs, follicles, ovary etc. Some figures lack of this information.
Response: Thank you for your comments. To facilitate reader understanding, we have edited the figure legends.
Why did the authors decide to perform an analysis of the effect of 5-HT on meiosis when they are interested in evaluating the role of 5-HT on follicle development?
What is the rationale for using the concentrations in Figure 3? Concentrations seem pretty high keeping in mind the ones used in figure 4. Can the decreased polar body extrusion at the highest concentration be due to cytotoxicity? Please clarify.
Response: Thank you for your comments. We would like to response the two questions together by discussing the progression of our initial experimental design. Our lab primarily focuses on research related to embryonic development and epigenetics. Literature reports have shown that serotonin (5-HT) is well-known for its role as a neurotransmitter, but it also functions as a developmental factor during embryogenesis, a topic that has been less explored. Therefore, we aimed to investigate whether serotonin affects porcine embryonic development and used mice as a model to explore the impact of 5-HT on embryonic development.
Our experiments revealed that serotonin indeed affects both porcine and murine embryonic development [1, 2]. It is understood that 5-HT is transported from the maternal system to the fetus via the SERT (serotonin membrane transporter) for embryonic development. This led us to want to know whether maternal 5-HT could influence oocyte maturation or follicular development. This line of inquiry formed the basis of the current study.
Thus, the concentrations of 500 μM and 1000 μM 5-HT used in the article were selected based on concentrations screened in embryonic studies and directly applied to oocytes to determine whether 5-HT affects oocyte maturation. If an effect was observed, it would justify further research into the role of 5-HT in follicular development. In our embryonic experiments, we verified that 500 μM and 1000 μM of 5-HT were non-toxic to embryos, which is why we did not conduct toxicity tests in this context. The results showed that at a concentration of 1000 μM, the first polar body extrusion was inhibited, indicating that meiosis did not occur and oocyte maturation was suppressed. These findings suggest that it would be meaningful to explore whether 5-HT also affects follicular development.
When studying the effects of serotonin on follicular GCs, we used concentrations referenced from the literature [3,4], as the literature had already tested for cytotoxicity. Therefore, we did not perform additional cytotoxicity tests in this context.
References:
[1] Han Y, Zhang M, Duan J, Zhang X, LI Z, Tang B. Effects of 5-hydroxytryptamine on porcine preimplantation embryo development. Chinese Journal of Veterinary Medicine, 2021, 41 (6): 1172-1178, 1186. In Chinese.
[2] Han Y, Zhang M, Duan J, Li L, Du J, Cheng H, Zhang S, Zhai Y, An X, Li Q, Zhang X, Li Z, Tang B. Maternal Prepregnancy 5-Hydroxytryptamine Exposure Affects the Early Development of the Fetus. Front Physiol. 2022; 13: 761357. doi: 10.3389/fphys.2022.761357.
[3] Gustafsson BI, Thommesen L, Stunes AK, Tommeras K, Westbroek I, Waldum HL, Slørdahl K, Tamburstuen MV, Reseland JE, Syversen U. Serotonin and fluoxetine modulate bone cell function in vitro. J Cell Biochem. 2006 May 1;98(1):139-51. doi: 10.1002/jcb.20734.
[4] John NJ, Lew GM, Goya L, Timiras PS. Effects of serotonin on tyrosine hydroxylase and tau protein in a human neuroblastoma cell line. Adv Exp Med Biol. 1991; 296:69-80. doi: 10.1007/978-1-4684-8047-4_8.
Minor comments:
- Line 40. Define GCs
Response: Thank you for your comments. We have added this information in the revised manuscript.
- Line 46. Which 3?
Response: Thank you for your suggestion .We have replaced “Estradiol is the representative steroid hormone among the three estrogens“with”Among estradiol, estrone and estriol, estradiol is the representative steroid hormone”.
- Line 75. On porcine cells…or in follicle development as that is what the authors stated they are interested in?
Response: Thank you for your comments.We are interested in the development of porcine follicular, and we have changed “The results may provide new insights into the mechanism of action of 5-HT on porcine follicles.”to“The results may provide new insights into the mechanism of action of 5-HT on porcine follicular development.”in the revised manuscript.
- Line 77. Results section is duplicated
Response: Thank you so much for your efforts to improve the quality of our manuscript.We have examined the whole manuscript, and deleted and revised the parts of discussion and result duplication. Delete the section 2.2 first sentence “5 - HT always works by binding to its receptors”.In the second paragraph of 3.Discussion, the last sentence of “which reveals that 5-HT receptors may have a potential impact on follicles development and 5-HT binds to different receptors and exerts different effects.” is replaced by “These findings suggest that serotonin receptors may potentially impact follicle development by binding to different receptors and exerting diverse effects.” Removed “GCs was the largest cell population in mature follicles, and participated in the regulation of follicular growth and development.” from 3. Discussion .
- Line 79. Verify? That means there is already a correlation established and you are just confirming it? Can the authors please provide the reference or rewrite the sentence?
This reviewer does not see any relevance for Table 2.
Response: Thank you for your comments. We have replaced “Verify” with “explore”. In this section we insert “The criteria for follicle grading are shown in Table 1”.
- Line 82. “which was the lowest in the whole follicle” What does it mean?
Response: Thank you for your comments. We have replaced “which was the lowest in the whole follicle” with “which was the lowest among the three sizes of follicles ”.
- Line 94. No significant difference in the size of follicles? Are the authors indicating that they analyzed size differences in the follicles using statistical analysis or that there was no difference in gene expression levels in the different follicles (based on the size)?
Response: Thank you for your comments. We have replaced “ there was no significant difference in the size of follicles ” with “ the expression of these genes was not significantly different in follicles of different sizes. ”
- Line 97. 5-HT and 5-HT1A… is it correct? Did the authors quantify the expression of 5-HT or the levels?
How long the COCs were exposed to 5HT? really important information is missing throughout the manuscript.
Response: Thank you for your comments. We quantified the expression of 5-HT in Fig1A. COCs were exposed to 5-HT for 48h. This information is contained in line 107 of the previous manuscript and in line 126 of the revised manuscript.
- Line 114. Should the sentence be in the past tense?
GCs and pGC?
Response: Thank you for your comments. We have modified the tense of this sentence. It should be pGCs, which we have modified uniformly.
- Line 119. Is “improved” the appropriate word to use?
Response: Thank you for your efforts to improve the quality of the manuscript. We have replaced “improved” with “increased. ”
- Line 190. 5-H2A?
Response: Thank you for your comments. We have replaced “5-H2A” with “5-HT2A” .
- Line 201. Did the authors evaluate ovulation?
Response: Thank you for your comments. We have replaced “ovulation” with “of the discharge of the first polar body of the oocyte. ”
- Lines 204-205 I do not believe it is possible to use GCs to investigate the regulatory mechanisms of 5-HT on follicle growth. Please, clarify
Response: Thank you for your suggestion to the article. GCs was the largest cell population in mature follicles, and participated in the regulation of follicular growth and development. Follicular granulosa cells promote the growth and differentiation of theca cells [1]. Energy substrates, nuclei required for oocyte metabolism Glycosidic acids and amino acids are supplied by granule cells through gap junctions [2]. Granulosa cells play a key role in follicular development and ovulation [3]. Granulosa cell apoptosis is the initiating signal for follicular atresia [4]. Based on the above literature review, we think it is possible to use GCs to investigate the regulatory mechanisms of 5-HT on follicle growth.
References:
[1]Nilsson E, Parrott JA, Skinner MK. Basic fibroblast growth factor induces primordial follicle development and initiates folliculogenesis. Mol Cell Endocrinol. 2001 Apr 25;175(1-2):123-30. doi: 10.1016/s0303-7207(01)00391-4.
[2]LAZZARI G, GALLI C, MOOR R M. Functional changes in the somatic and germinal compartments during follicle growth in pigs . Animal Reproduction Science, 1994, 35(1-2):119-30.
[3]Braw-Tal R. The initiation of follicle growth: the oocyte or the somatic cells? Mol Cell Endocrinol. 2002 Feb 22;187(1-2):11-8. doi: 10.1016/s0303-7207(01)00699-2.
[4]Zheng Y, Ma L, Liu N, Tang X, Guo S, Zhang B, Jiang Z. Autophagy and Apoptosis of Porcine Ovarian Granulosa Cells During Follicular Development. Animals (Basel). 2019 Dec 10;9(12) doi:10.3390/ani9121111.
- Line 257. How did the authors perform the follicle separation and measure of the follicles?
Response: Thank you for your comments. The pig ovaries were washed 3 times with PBS at 37℃, and the attached tissues such as the ovarian frenum were removed, then immersed in 75% alcohol for 30s. After being washed once with PBS to remove the alcohol, the ovaries were placed in PBS and kept warm in a 37℃ water bath. A small number of ovaries were removed and placed in a sterile Petri dish covered with PBS, and individual follicles were dissected with the use of autoclaved scissors and forceps. Follicles were classified into three size categories using calliper. We have added this part to the materials and methods section in the revised manuscript.
- Line 258. Table 1?
Response: Thank you for your comments. We have corrected it.
- Line 261. Did the authors wash the “follicular fluid” or the cells?
Response: Thank you for your comments. We have replaced “follicular fluid was washed in PBS solution several times” with “the supernatant was decanted and the cell precipitate was washed several times in PBS solution” .
- Line 266. Where does this RNA come from? It is not clear what tissue/follicle population the authors used. Please clarify.
Response: Thank you for your comments. GCs, follicles, and ovarian tissue RNA were isolated using Trizol. We have inserted this sentence into the manuscript.
Sections in the materials and methods can be rearranged to have a logical sequence.
Response: Thank you for your suggestion to the article. Sections in materials and methods have been rearranged.
- Line 272. Rohche?
Response: Thank you for your comments. We have corrected it to “Roche”.
- Line 276. 1x103? Superscript
Response: Thank you for your comments. We have corrected it.
- Line 278. Treated with the fixation… fixed
Response: Thank you for your comments.We have changed “fixation” to “fixed”.
Section 4.6 are the authors using a commercial kit? The authors need to clarify what are the reagents B, C, D, and E.
Response: Thank you for your comments. Yes, we used the kit, which has been explained in section 4.6 of the revised manuscript.
Comments on the Quality of English Language
English editing is needed as the document is poorly written with numerous grammatical and writing errors. It is hard to follow and confusing. Some examples are in the below.
Response: We have thoroughly checked and revised the writing and grammar of the manuscript.
- Line 61. Folliculars
Response: Thank you for your comments. We have changed folliculars to follicular.
- Line 78. Different degrees of follicles?
Response: Thank you for your efforts to improve the quality of the manuscript. We have changed “different degrees of follicles” to “follicles of different sizes”.
- Line 108. Ma turation
Response: Sorry, We made a mistake. We have corrected it.
Round 2
Reviewer 2 Report
Comments and Suggestions for Authors
Thank you for considering my suggestions.
After reading the new version of the manuscript, I can see that it has improved.
My only doubt is the answer to the last point regarding cell proliferation. While the level of cyclins is a certain indicator of proliferative capacity, there is a lack of cytometric methods that would unequivocally demonstrate the level of cell proliferation increase. Currently, the conclusion about the influence of serotonin on proliferation is inadequate in relation to the research methods used.
Author Response
After reading the new version of the manuscript, I can see that it has improved.
My only doubt is the answer to the last point regarding cell proliferation. While the level of cyclins is a certain indicator of proliferative capacity, there is a lack of cytometric methods that would unequivocally demonstrate the level of cell proliferation increase. Currently, the conclusion about the influence of serotonin on proliferation is inadequate in relation to the research methods used.
Response: Thank you for your comments. I’m sorry we couldn’t answer your question clearly. As you suggested, we counted the positive cells: there were 13 positive cells in the control group, with a total cell count of 215, while the 5-HT experimental group had 45 positive cells out of a total of 319. The positive cell rate of the control group was 6.05%, while that of the 5-HT group was 14.11%. The chi-square test indicated that the difference between the two groups was significant, demonstrating that 5-HT indeed promotes significant cell proliferation in GCs. The results have been incorporated into result Subsection 2.4 and the figure legend for Figure 4, specifically in lines 155-160 and lines 163-168 in the revised manuscript.
Lines 155-160 of the manuscript are as follows:
“Additionally, the EdU results indicated that the positive cell rate of the control group was 6.05%, while that of the 5-HT group was 14.11%. This implies that 5-HT facilitated cell proliferation (Fig. 4B). These findings suggest that 5-HT promotes the proliferation of porcine granulosa cells, indicating its potential role in enhancing follicular growth and development through the regulation of cell cycle progression.”
lines 163-168 of the manuscript are as follows:
“chi-square test was conducted to evaluate the statistical difference in the proportion of positive cells between the control group (n = 215, positive = 13, negative = 202) and the experimental group (n = 317, positive = 45, negative = 272). The test revealed a significant difference (χ² = 85.43, df = 1, p < 0.05), indicating that the proportion of positive cells varied significantly between the two groups. NC: Control group; 5-HT: 5-HT group.”
Reviewer 3 Report
Comments and Suggestions for Authors
Although the manuscript has been improved, this reviewer believes that the manuscript needs to be improved before acceptance. Some of the main concerns from the last report were not properly addressed.
Comments on the Quality of English LanguageEnglish editing is still needed. Some of the added sentences are examples of it.
Author Response
Although the manuscript has been improved, this reviewer believes that the manuscript needs to be improved before acceptance. Some of the main concerns from the last report were not properly addressed.
English editing is still needed. Some of the added sentences are examples of it.
Response: Thank you for your comments.
1.We have reorganized the introduction, with the first paragraph focusing on the follicle and its importance. The second and third paragraphs describe the roles of GCs and E2 in the follicle, while the fourth paragraph discusses 5-HT and its possible effects on GCs and E2.2.We checked
2.the figure notes again and added more detailed information.
3.Once again, we reviewed the full text for English editing and made revisions.
Your comments are very valuable, and we have made significant efforts to address them. We have done our best to revise the language in order to enhance the overall quality of the article and meet your expectations. We apologize for any remaining shortcomings in our article. In the future, when we have sufficient funding, time, and personnel, we will further enhance the research content and continue our in-depth studies. We sincerely appreciate the reviewers' suggestions once again.